# High Genetic Diversity and No Evidence of Clonal Relation in Synchronous Thyroid Carcinomas Associated with Hashimoto’s Thyroiditis: A Next-Generation Sequencing Analysis

**DOI:** 10.3390/diagnostics10010048

**Published:** 2020-01-17

**Authors:** Csaba Molnár, Emese Sarolta Bádon, Attila Mokánszki, Anikó Mónus, Lívia Beke, Ferenc Győry, Endre Nagy, Gábor Méhes

**Affiliations:** 1Department of Pathology, Faculty of Medicine, University of Debrecen, H-4032 Debrecen, Hungary; molnar.csaba@med.unideb.hu (C.M.); badon.emese.sarolta@med.unideb.hu (E.S.B.); mokanszki.attila@med.unideb.hu (A.M.); monusne.aniko@med.unideb.hu (A.M.); beke.livia@med.unideb.hu (L.B.); 2Department of Surgery, Faculty of Medicine, University of Debrecen, H-4032 Debrecen, Hungary; gyory@med.unideb.hu; 3Division of Endocrinology, Department of Internal Medicine, University of Debrecen, H-4032 Debrecen, Hungary; endre.nagy@med.unideb.hu

**Keywords:** autoimmune thyroiditis, mutational profiling, papillary thyroid carcinoma, mutagenesis, next-generation sequencing (NGS), variant allele frequencies (VAF)

## Abstract

The close association between pre-existing Hashimoto’s thyroiditis and thyroid cancer is well established. The simultaneous occurrence of multiple neoplastic foci within the same organ suggests a common genotoxic effect potentially contributing to carcinogenesis, the nature of which is still not clear. Next-generation sequencing (NGS) provides a potent tool to demonstrate and compare the mutational profile of the independent neoplastic foci. Our collection of 47 cases with thyroid carcinoma and Hashimoto’s thyroiditis included 14 with at least two tumorous foci. Detailed histological analysis highlighted differences in histomorphology, immunoprofile, and biological characteristics. Further, a 67-gene NGS panel was applied to demonstrate the mutational diversity of the synchronic tumors. Significant differences could be detected with a wide spectrum of pathogenic gene variants involved (ranging between 5 and 18, cutoff >5.0 variant allele frequencies (VAF)). Identical gene variants represented in both synchronous tumors of the same thyroid gland were found in only two cases (*BRAF* and *JAK3* genes). An additional set of major driver mutations was identified at variable allele frequencies in a highly individual setup suggesting a clear clonal independence. The different *BRAF* statuses in coincident thyroid carcinoma foci within the same organ outline a special challenge for molecular follow-up and therapeutic decision-making.

## 1. Introduction

Thyroid carcinoma is the most frequent endocrine neoplasia, the incidence of which is increasing worldwide [1,2]. Major etiological factors include environmental and direct thyrotoxic effects, and selected comorbidities such as autoimmune thyroid disease are also considered as associated factors [3,4]. While the immunological background and pathological consequences of autoimmune lymphocytic thyroiditis and, more recently, IgG4-related chronic inflammation have been gradually elucidated, the exact mechanisms of subsequent carcinogenesis are mostly unknown. Hashimoto’s thyroiditis (HT) is one of the most frequent autoimmune thyroid disorders characterized by massive lymphocytic infiltrate and progressive destruction of the thyroid parenchyma [5,6]. The autoimmune attack results in direct cellular toxicity, and regeneration is associated with endocrine and metabolic/oxidative stress at the level of the thyroid follicular epithelium [7,8]. Parenchymal hypofunction, on the other hand, triggers a constant proliferative stimulus by thyroid-stimulating hormone (TSH), enabling increased survival and adaptive selection, which are both important factors in the process of carcinogenesis. According to the current view, not only has the association between HT and especially papillary-type thyroid carcinoma (PTC) been established, but the frequent multifocality of PTC can also be explained by HT-related injury and consequent stimulatory mechanisms [6,9].

With the recent developments of genomic DNA testing, a sharpened picture of relevant gene alterations and molecular pathways can also be provided for thyroid malignancies. The availability of next-generation sequencing (NGS) approaches enables the large-scale analysis of cancer genotype from archived pathological material [10,11]. Several well-documented studies using NGS were performed to define the mutational spectrum of thyroid carcinoma, especially focusing on advanced and anaplastic carcinoma cohorts [12,13]. Data obtained by extended cancer-related gene panel analysis delineated MAPK and PI3K pathway mutations as contributing to aggressive behavior. Following clinical interpretation of sequencing results, a path was cleared for targeted therapeutic suggestions [14,15,16]. On the other hand, there is only limited information regarding the origin and genetic background of thyroid cancer (TC) associated with HT. Interestingly, *BRAF* V600 mutations were under-represented in association with HT-related PTC; however, other characteristic changes or frequent gene variant patterns could not be identified [9]. Findings on multiple neoplastic nodules and their clonal relation in the context of HT have also not been reported in the literature so far. Therefore, following the histopathological evaluation of multifocal thyroid lesions provided in our single-center cohort, we performed an NGS-based 67 multigene analysis in a selected set of HT-related TC cases to highlight the mutational pattern and potential common features of this special and complex form of carcinogenesis.

## 2. Materials and Methods

### 2.1. Tissue Samples and Histological Workup

Histological sections and clinicopathological data were used of TC patients operated on at the Department of Surgery, University of Debrecen between 2007 and 2015 and diagnosed with the parallel occurrence of TC and HT. Ethical aspects were covered by the approval of the Hungarian National Health Science Committee (reg. no. 60355-2/2016/EKU). Histological slides were rescreened for tumor multifocality and presence of lymphocytic infiltrate. HT was suggested if massive chronic lymphocytic infiltration, secondary lymphatic follicles, and atrophy of the thyroid follicular epithelium were obvious. The clinical history together with radiological and serological findings further supported the autoimmune background and the diagnosis of HT in these cases. TC was confirmed by classical histological criteria (cyto- and nuclear morphology, including nuclear clearing, irregularity, grooving, and pseudoinclusions, as well as psammoma bodies) in hematoxylin-eosin-stained conventional sections. Multifocality of TC was stated when more than one completely separate tumor cluster was found at a distance of larger than 5 mm during the review of the surgical resection material. Immunohistochemistry (IHC) for tissue antigens HBME1, galectin-3, CD56, and CK19 was applied to clearly demonstrate the extent of thyroid cancer within individual tumor foci. IHC was done according to our routine diagnostic protocols in a Ventana Ultra stainer (Roche Diagnostics Mannheim, Germany). In addition, mutant *BRAF* status (clone VE1), the p53 status (clone DO-7), and the cell proliferation rate (clone Mib-1) was also determined. Individual TC foci were compared for IHC characteristics by conventional light microscopy by two experienced pathologists (C.M. and G.M.).

### 2.2. DNA Isolation

DNA from the individual TC foci was removed and processed from the formaldehyde fixed paraffin embedded (FFPE) sections following accurate dissection of the tumor area using a standard protocol. Genomic DNA was extracted using the QIAamp DNA FFPE Tissue Kit (Qiagen, Hilden, Germany). DNA concentration was measured with the Qubit dsDNA HS Assay Kit in a Qubit 4.0 fluorometer (Thermo Fisher Scientific, Waltham, MA, USA).

### 2.3. BRAF Mutation Testing

*BRAF* testing focusing on the clinically relevant tyrosine kinase domain mutation status was independently performed in a routine session using conventional Sanger sequencing based on Big Dye chemistry (Applied Biosystems, Foster City, CA, USA) in an ABI 310 genetic analyzer.

### 2.4. Library Preparation for NGS

The amount of amplifiable DNA (ng) was calculated according to the Archer PreSeq DNA Calculator Assay Protocol (Archer DX, Boulder, CO, USA). After fragmentation of the genomic DNA, libraries were created by the Archer VariantPlex Solid Tumor Kit (Archer DX, Boulder, CO, USA). The solid tumor panel included the following 67 genes: *ABL1*, *AKT1*, *ALK*, *APC*, *ATM*, *AURKA*, *BRAF*, *CCND1*, *CCNE1*, *CDH1*, *CDK4*, *CDKN2A*, *CSF1R*, *CTNNB1*, *DDR2*, *EGFR*, *ERBB2*, *ERBB3*, *ERBB4*, *ESR1*, *EZH2*, *FBXW7*, *FGFR1*, *FGFR2*, *FGFR3*, *FLT3*, *FOXL2*, *GNA11*, *GNAQ*, *GNAS*, *H3F3A*, *HNF1A*, *HRAS*, *IDH1*, *IDH2*, *JAK2*, *JAK3*, *KDR*, *KIT*, *KRAS*, *MAP2K1*, *MDM2*, *MET*, *MYC*, *MYCN*, *MLH1*, *MPL*, *NOTCH1*, *NPM1*, *NRAS*, *PDGFRA*, *PIK3CA*, *PIK3R1*, *PTEN*, *PTPN11*, *RB1*, *RET*, *RHOA*, *ROS1*, *SMAD4*, *SMARCB1*, *SMO*, *SRC*, *STK11*, *TERT*, *TP53*, and *VHL*. The KAPA Universal Library Quantification Kit (Kapa Biosystems, Roche, Basel, Switzerland) was used for the final quantification of the libraries. 

### 2.5. Next-Generation Sequencing

The MiSeq System (MiSeq Reagent kit v3 2 × 300 cycles, Illumina, San Diego, CA, USA) was used for sequencing. The libraries (final concentration of 4 nM, pooled by equal molarity) were denatured by adding 0.2 nM NaOH and diluted to 40 pM with hybridization buffer from Illumina. The final loading concentration was 14 pM libraries and 5% PhiX. Sequencing was conducted according to the MiSeq instruction manual.

### 2.6. Data Analysis

Raw sequence data were analyzed with Archer analysis software (version 6.2.; Archer DX, Boulder, CO, USA) for the presence of single-nucleotide variants (SNVs) as well as insertions and deletions (indels) [17]. For the alignment, the human reference genome GRCh37 (equivalent UCSC version hg19) was built. Molecular barcode (MBC) adapters were used to count unique molecules and characterize sequencer noise, revealing mutations below standard NGS-based detection thresholds. The sequence quality for each sample was assessed and the cutoff was set to 5% variant allele fraction. Total depth of sequencing required 20.9 million reads. A reliable variant detection required a coverage of >250 reads. The limit of detection of this panel was 3 detected variants per 250 reads (the so-called alternate observation). Large insertion/deletion (>50 bp) and complex structural changes could not be captured by the method. The results were described using the latest version of Human Genome Variation Society (HGVS) nomenclature for either the nucleotide or protein level (http://varnomen.hgvs.org/). Individual gene variants were cross-checked in the Cosmic v90 database (Catalogue of Somatic Mutations in Cancer, available online: http://cancer.sanger.ac.uk/cosmic, accessed on 5 September 2019) for clinical relevance.

## 3. Results

Out of the 262 total TC cases evaluated, 47 were included based on clinical history and morphology consistent with autoimmune thyroiditis. Multiple tumor formation was demonstrated in 14 cases (29.8%). The lymphocytic infiltrate of the non-neoplastic parenchyma was significant and presented with slightly variable (medium to high) intensities in and around the coincidental tumors. The independent carcinomatous foci were identified with variable sizes (2–23 mm) and frequently with different histological and biological characteristics. Histopathological and biological features of all 14 matched duplicate thyroid carcinomas are displayed in Table 1. Also, the histological subtype most frequently fitting the criteria of classical papillary (cP, 13 samples) or papillary follicular variant (PF, 7 samples), diffuse sclerotizing variant (5 samples), and oncocytic histological variant (3 samples) were diagnosed. Discrepancies among the histological subtypes between parallel tumor foci could be identified in 6/14 cases. Coincidence was observed of cP and oncocytic variant in two cases, cP and diffuse sclerotizing variant in three cases, and oncocytic and PF variant in one case. The cell proliferation rate determined by the Mib1 IHC labeling index was generally low and did not differ significantly between matched tumor foci. The IHC staining result for the mutant *BRAF* protein (VE1) was comparable in all but one case. Simultaneous *BRAF* −/− samples were seen in 9/14 (64.3%), *BRAF* +/+ samples in 4/14 (28.6%) cases, while a discordant *BRAF* +/− pattern was demonstrated in 1/14 (7.1%) cases. The tumor cell proliferation was generally low, with only <2% Mib-1 labeling in all of the evaluated samples, irrespective of tumor stage, size, histological variant, and *BRAF* status. H-scores for nuclear p53 immunostaining proved to be variable but generally of weak/intermediate level (range of 0–160), which was not consistent with a mutant-type p53 pattern (Table 1). The two independent foci of simultaneous TC pairs from six cases were further sequenced using the VariantPlex 67-gene NGS panel (Table 2).

The 12 DNA samples isolated from macrodissected FFPE tissue material corresponded to tumor cell contents of over 70%, and all quantitative and qualitative requirements appeared optimal for library preparation and sequencing.

The 67-gene panel data from the sequenced 12 TC tumor DNA resulted in a large number of exonic nucleotide variants in highly variable amounts (13–111 SNVs/sample). The vast majority represented missense variants associated with amino acid exchange (range of 7–65 variants/samples). Altogether, 41 pathogen gene variants were identified at the level of 5% variant allele frequencies (VAF) (0.05) presenting with a highly variable rate and composition (Figure 1).

The involvement of the *APC* gene was observed most frequently (16 variants, also in multiple fashions), followed by *DDR2* (8 variants) and *FBXW7* (6 variants). Both the distribution and the allele frequency (VAF) of the variants detected within the individual TC tumors were highly different. The number of coding gene variants ranged between 3 and 18 per sample, with the highest allele frequency of 0.447 (*JAK3* variant in case 5). *BRAF* pathogenic variants clearly validated the IHC positivity obtained by the VE1 clone against the mutant protein in all three affected samples. *TP53* variants were under-represented and generally present at only low frequencies (VAF range of 0.06–0.11). The presence of *TP53* variants was not associated with special phenotypic features and low allele frequencies were consistent with the low/intermediate H-scores obtained following IHC using the anti-p53 antibody DO-7.

Table 2 shows the data obtained after the comparison of coincident tumor foci within the same thyroid gland. The number of the matching gene variants was surprisingly low. Only two out of six cases presented with overlapping variant profiles, which were limited to a single event in both cases. In case 1, we identified the same *BRAF* V600E mutation (*BRAF*: c.1799T>A, p.Val600Glu) in both samples with VAF values of 0.235 and 0.097. The identical *JAK3* variant (*JAK3*: c.2164G>A, p.Val722Ile) could be demonstrated in case 5 at high frequencies (VAF of 0.30 and 0.44) in the parallel tumors. In the remaining four out of six cases, no single gene variant overlapped within the simultaneous tumor genomes despite the relatively large series of positive events, including characteristic driver mutations (e.g., *VHL*, *KRAS*, *PIK3CA*, *PTEN*, and *TP53*). Interestingly, the allele frequencies of these individual gene variants were highly variable, and the majority of them presented with relatively low frequencies, suggesting a potential subclonal, passenger nature (Figure 2).

## 4. Discussion

NGS is a unique tool that can demonstrate the complex mutational profile in malignancies from clinical samples and also enables the analysis of DNA from archived FFPE tumor tissues, provided that the tissue sample is representative and the quality of the isolated genomic DNA is preserved [18,19]. Both diagnostic and predictive indications have been increasingly reported, and NGS-based thyroid cancer testing has become generally available. Gene panel sequencing has recently been considered to differentiate thyroid nodules with indeterminate cytological findings, which may support decisions for surgery [16,20]. On the other hand, NGS was found to be highly effective at defining targetable molecular pathways in advanced as well as poorly differentiated/anaplastic thyroid carcinomas [10,12,14,21]. Moreover, multigene DNA sequencing was reported to be useful for differentiating PTC developing as part of familial cancer syndromes at young ages, as was presented, for example, for familial adenomatous polyposis (FAP) syndrome [22].

In addition to what was previously mentioned, NGS-based gene panel testing enables the performance of comparative studies and highlights the potential clonal relation of simultaneously occurring malignancies. This is particularly relevant for multiple independent tumors developing in the same organ suggesting some principal common pathobiology. To date, multiple neoplasias of the thyroid were only compared at a single-case level by detailed mutational testing, according to our knowledge [23,24].

In the current study, 14 TC cases were selected for multifocality and unambiguous HT etiology. The individual tumors presented with significant diversity regarding histological subtype, immunophenotype, and *BRAF* status, at least to some extent. However, features potentially associated with the HT etiology were not obvious. We assume that the described high pathomorphological variability is the result of complex processes determined by individual genetic/epigenetic changes. To obtain greater insight into these features, NGS sequencing data of six available cases with double malignancies were processed and compared.

The VariantPlex solid tumor NGS panel used in our study simultaneously detected and characterized SNVs and indels in 67 genes commonly associated with solid tumor progression. On the other hand, this panel was not designed to detect large sequential alterations and structural rearrangements of the genome. This restriction should be considered, especially as chromosomal translocations (e.g., RET/PTC) appear to be clinically significant in thyroid tumors, primarily in the classical papillary variant [25]. In the present study, however, we were principally interested in SNVs in HT-related carcinoma and, further, in the potential clonal correlations characteristic of multifocality. Using the VariantPlex panel, we identified pathogenic variants of 41 genes in total, which were described in association with cancer according to the Cosmic database. Several genes harbored multiple variants. Interestingly, the involvement of the *APC* gene was the most frequent in our cohort of 12 TCs associated with HT. However, comparison of the exact SNV sites of the simultaneous nodules did not reveal any exact match of the detected *APC* alterations. Therefore, the individual changes were interpreted as tumor-related somatic variants, and the germline involvement of the *APC* gene in the evaluated patients could be excluded. Further frequent variants affected *DDR2*, *FBXW7*, *CDH1*, *NOTCH1*, and *VHL* genes. Interestingly, the dedicated members of the classical MAPK pathway (*EGFR1*, *KRAS*, *NRAS*, *BRAF*, *RET*, etc.) were generally under-represented. Due to the high number and the distinct distribution of variants, the mutational changes of the HT-related TCs generally appeared to deviate from the gene involvement described for classical papillary TCs [26,27].

The comparison of coincidental tumors from the same thyroid gland resulted in highly divergent genetic profiles in the individual TCs. Common gene variants could be demonstrated in only two out of the six evaluated cases, each presenting with a single gene variant occurring in both tumor foci (*BRAF* and *JAK3*). However, each tumor pair featured several additional dominant variants which were unrelated. The specific *JAK3* variant c.2164G>A (p.Val722Ile) identified in case 11 is unique and, thus, would suggest a common origin of the two neoplastic foci. As such, this would be the only sign of an early genetic hit, while additional changes indicate independent tumor development and do not support the potential of intraorgan metastatic spread. On the other hand, *BRAF* c.1799T>A (p.Val600Glu, referred to as V600E in the literature) kinase domain mutations are frequent enough in TC to occur incidentally in both coexisting foci, as we found in case 1. The value of the *BRAF* V600E variant as a clonality marker is therefore questionable in this particular case. Apart from these exceptions, we could not provide any further trace of genetic relationship between independent foci from the same thyroid gland. 

Our result on multiple TCs associated with HT is based on low case numbers and should be considered as preliminary. Previous findings on multifocal carcinogenesis are rare but all support the idea of independent clonal evolution of simultaneous tumors within the same thyroid tissue. Whole exome and transcriptome sequencing provided evidence that coincidental adenomatoid nodules and a PTC from the same surgical sample had no overlapping mutations [23]. Similar conclusions were made in the report on a case with simultaneous independent papillary and medullary carcinomas [24]. Moreover, distinct morphological components of a composite thyroid carcinoma also had highly different mutational profiles, indicating divergent growth and no clonal association despite the close spatial relationship [28].

According to earlier data, multifocality is a hallmark of radiation-related TCs, which are characterized by genetic diversity and inconsistent gene variant patterns [29,30]. Similar to radiotoxic injury, thyroid follicular cells are exposed to continuous and direct cellular irritation as a consequence of autoimmune inflammation. Oxidative stress and peroxidation are suggested to be major factors in thyroid carcinogenesis, as the deregulation of H_2_O_2_-dependent thyroid hormone synthesis in damaged follicular epithelial cells contributes to massive DNA damage [31,32]. In line with this, a clear increase in thyroperoxidase expression and function as well as the reduction of the neutralizing GSH activity in association with manifest HT in the thyroid parenchyma was reported [33]. As such, an immune attack associated with oxidative stress may be a key mechanism causing enhanced random DNA damage throughout the organ in HT, resulting in multiple transformation foci characterized by numerous and highly variable mutations.

Current data suggest characteristic genetic patterns for different thyroid carcinoma etiologies. Sporadic, classical forms appear to be associated with isolated defects of the MAPK pathway and frequent *BRAF* mutations. Early onset multifocal PTC may carry the *APC* gene mutation and, as such, falls under the familial cancer syndrome FAP. Radiation-related TC differs from the sporadic forms escaping the classical molecular pathways and presenting with highly variable mutational profiles. Similar to this, we now provide data indicating the high genetic diversity and a multiclonal tumorigenesis in relation with clinically significant autoimmune thyroid disease. As demonstrated by our comparative NGS studies, the genetic diversity may also lead to discrepancies of clinically relevant mutations (e.g., *BRAF* and *NRAS*). Multiple lesions with different mutational profiles define a new challenge for therapeutic decision-making and long-term follow-up of patients with thyroid cancer.

## Figures and Tables

**Figure 1 diagnostics-10-00048-f001:**
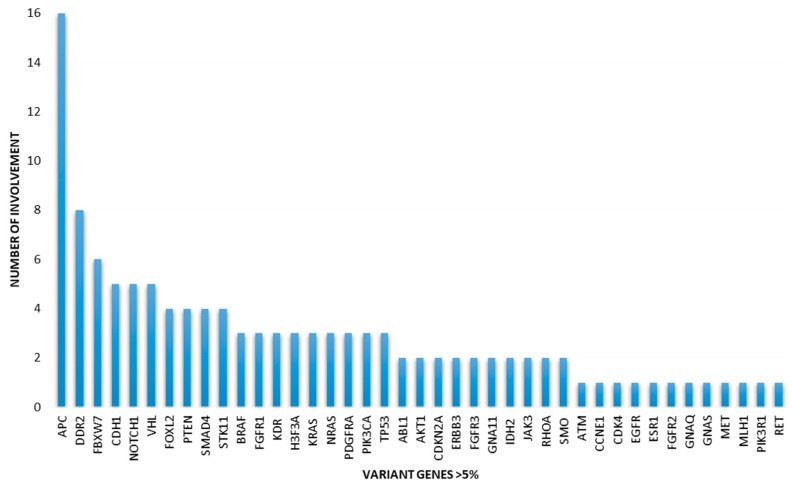
The spectrum and frequency of pathogenic gene variants identified in synchronous thyroid carcinomas associated with Hashimoto’s thyroiditis. (*APC*: adenomatous polyposis coli; *DDR2*: discoidin domain receptor tyrosine kinase 2; *FBXW7*: F-box/WD repeat-containing protein 7; *CDH1*: cadherin 1; *NOTCH1*: Notch homolog 1, translocation associated; *VHL*: von Hippel-Lindau tumor suppressor; *FOXL2*: forkhead box protein L2; *PTEN*: phosphatase and tensin homolog; *SMAD4*: SMAD family member 4; *STK11*: serine/threonine kinase 11; *BRAF*: B-Raf proto-oncogene; *FGFR1*: fibroblast growth factor receptor 1; *KDR*: kinase insert domain receptor; *H3F3A*: H3.3 histone A; *KRAS*: Kirsten Rat Sarcoma Viral Oncogene Homolog; *NRAS*: neuroblastoma RAS Viral Oncogene Homolog; *PDGFRA*: platelet-derived growth factor receptor alpha; *PIK3CA*: phosphatidylinositol-4,5-bisphosphate 3-kinase catalytic subunit alpha; *TP53*: tumor protein p53; *ABL1*: ABL proto-oncogene 1, nonreceptor tyrosine kinase; *AKT1*: AKT serine/threonine kinase 1; *CDKN2A*: cyclin-dependent kinase inhibitor 2A; *ERBB3*: erb-b2 receptor tyrosine kinase 3; *FGFR3*: fibroblast growth factor receptor 3; *GNA11*: G protein subunit alpha 11; *IDH2*: isocitrate dehydrogenase 2; *JAK3*: Janus kinase 3; *RHOA*: Ras Homolog Family Member A; *SMO*: Smoothened, Frizzled Class Receptor; *ATM*: Ataxia Telangiectasia Mutated; *CCNE1*: Cyclin E1; *CDK4*: Cyclin-Dependent Kinase 4; *EGFR*: endothelial growth factor receptor; *ESR1*: Estrogen Receptor 1; *FGFR2*: Fibroblast Growth Factor Receptor 2; *GNAQ*: G Protein Subunit Alpha Q; *GNAS*: G Protein Subunit Alpha S; *MET*: Hepatocyte Growth Factor Receptor; *MLH1*: DNA Mismatch Repair Protein Mlh1; *PIK3R1*: Phosphoinositide-3-Kinase Regulatory Subunit 1; *RET*: Proto-Oncogene Tyrosine-Protein Kinase Receptor Ret).

**Figure 2 diagnostics-10-00048-f002:**
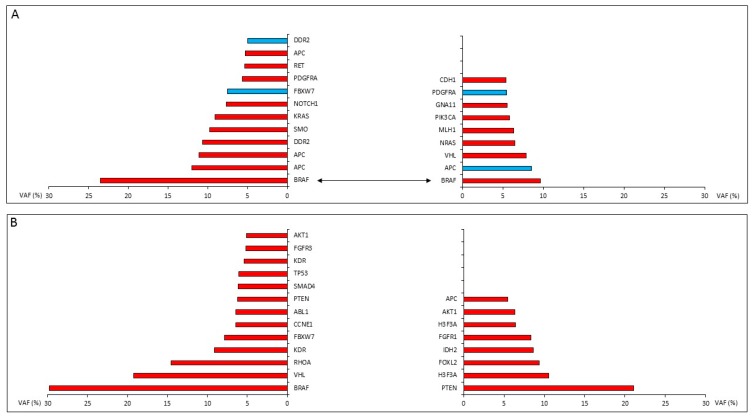
Results of the 67-gene NGS sequencing from two cases with synchronously growing thyroid carcinomas in association with Hashimoto’s thyroiditis. (**A**) Independent synchronous tumors from the same thyroid gland (case 1) shared the classical V600E mutant *BRAF* genotype but no other variants. (**B**) No single match within the series of 13 and 8 gene variants detected in the two simultaneously growing tumors of case 14. Red bars represent missense variants, and blue bars indicate stop gain. *x* axis: allele frequency of the gene variant (variant allele frequencies (VAF) %) identified.

**Table 1 diagnostics-10-00048-t001:** Multiple thyroid cancer (TC) tumor foci associated with Hashimoto’s thyroiditis (HT): histology and functional parameters.

	Case No.	Sample	Tumor Size (mm)	TNM Class	Histology Subtype	Grade of Inflammation	MIB1 IHC (%)	*BRAF* VE IHC
*BRAF* +/+	1 *	A	10	pT2	DS	m	<1%	pos
		B	6	pT2	cP	m	<1%	pos
	2	A	7	pT3	cP	h	<5%	pos
		B	5	pT3	cP	h	<5%	pos
	3	A	7	pT1b	DS	m	<5%	pos
		B	5	pT1b	cP	m	<5%	pos
	4	A	12	pT1b	DS	h	<5%	pos
		B	2	pT1b	cP	h	<5%	pos
*BRAF* −/−	5	A	5	pT1a	cP	m	<1%	neg
		B	3	pT1a	cP	m	<1%	neg
	6	A	4	pT1a	O	m	<5%	neg
		B	6	pT1a	cP	m	<5%	neg
	7	A	6	pT1a	cP	m	<5%	neg
		B	3	pT1a	cP	m	<5%	neg
	8 *	A	11	pT1a	cP	m	<1%	neg
		B	12	pT1a	O	m	<1%	neg
	9 *	A	10	pT1b	PF	h	<1%	neg
		B	6	pT1b	PF	m	<1%	neg
	10 *	A	11	pT1a	DS	h	<1%	neg
		B	8	pT1a	DS	h	<1%	neg
	11 *	A	13	pT1b	PF	h	1–2%	neg
		B	6	pT1b	O	h	<1%	neg
	12	A	5	pT1a	cP	m	<5%	neg
		B	4	pT1a	cP	m	<5%	neg
	13	A	4	pT1a	PF	h	<5%	neg
		B	2	pT1a	PF	h	<5%	neg
*BRAF* +/−	14 *	A	23	pT3	PF	m	1%	pos
		B	17	pT3	PF	m	<1%	neg

Samples A and B represent two individual neoplastic foci within the same thyroid gland (TNM: tumor, nodes, and metastases; histological subtypes: DS: diffuse sclerotic, cP: classical papillary, PF: papillary follicular variant, O: oncocytic; grade of inflammation: h: high, m: medium; IHC: immunohistochemistry). * sample tested by next-generation sequencing (NGS).

**Table 2 diagnostics-10-00048-t002:** Number of gene alterations detected and details of the identical variants in the matched TC determined by a 67 multigene NGS panel. Samples A and B refer to tumors presented in Table 1. (AF: allele frequency).

Case No.	Sample	Total no. of Variants	No. of Identical Variants	Identical Gene	Type of Identical Variants	AF of Identical Variants (%)
1	A	12	1	*BRAF*	p.Val600Glu (c.1799T>A)	23
	B	9				9
8	A	18	0	-	-	-
	B	5				-
9	A	9	0	-	-	-
	B	15				-
10	A	11	0	-	-	-
	B	10				-
11	A	3	1	*JAK3*	p.Val722Ile (c.2164G>A)	30
	B	7				44
14	A	13	0	-	-	-
	B	8				-

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
