# Peer review of "High Genetic Diversity and No Evidence of Clonal Relation in Synchronous Thyroid Carcinomas Associated with Hashimoto’s Thyroiditis: A Next-Generation Sequencing Analysis"

_diagnostics, 2020, doi:10.3390/diagnostics10010048_

Round 1
Reviewer 1 Report
The article titled: High genetic diversity and no evidence of clonal relation in synchronous thyroid carcinomas associated with Hashimoto's thyroiditis – a next-generation sequencing analysis contains important information helping genetic analysis of thyroid cancer. Authors have used NGS and meticulously analysed the data generated by the relevant software and showed the presence of numbers of genes associated with Hashimoto's thyroiditis. The abstract is concise and describe the methods and results. The introduction is reasonably well-written with correct referencing. The description of the methods is somehow weak and needs more detailed information, for example., in lines, 125 -126 "The mutant BRAF protein expression (VE1) was comparable in all but one cases." How did you measure the level of protein expression? Please explain and describe the NGS briefly and why this method is superior. Then other clinical methods used for the diagnosis of Hashimoto's and Thyroid cancer should be employed in routine diagnosis. The discussion was not sufficiently detailed, and authors should justify with references the applicability of the is genetic analysis in future diagnosis. The tables need legends, and all abbreviations need an explanation. Figure 1 and 2 are of significance; please write the full names of the variants genes. Expand the method section and detail the technologies used, provide comprehensive discussion to justify the use of NGS in future routine analysis. Authors should provide strength as well as the weakness of the study and provide a future direction. Understandably, the sample size is small and not always toxic nodules are present.
Author Response
Thank you for the supporting opinion and for the valuable remarks. Below please find our answers to the open issues and the modifications done:
The presence of mutant BRAF protein was determined by IHC using the VE1 antibody as usual. The result of the immunostaining was simply classified as negative/positive in the matched simultaneous thyroid carcinomas without further quantification as the expression level of the mutant protein was not the main focus of the current study. The sentence was clarified (line 144-145). The existance of HT in TC samples was first suggested by the presence of masses of lymphocytic infiltrate and reactive B-lymphoid germinal centers as expected. Careful analysis of patient histories in these cases revealed long term thyroid pathology with radiological iaming and serological findings supporting HT (line 73-75). References for NGS for clinical routine were included (Ref. no. 18 and 19) Strengths and limitations of NGS are now more clearly reflected: the ultimative strength of NGS-based gene panel testing is the simultaneous presentation of the complex mutational landscape with the ability to highlight the spectrum of clinically relevant actionable mutations or resistance mechanisms (lines 224-235) weaknesses include the limitation to analyze SNV/short sequence alterations without the opportunity to present structural aberrations, such as gene fusions or large deletions. Analysis of DNA derived from FFPE tumor material is indeed applicable by NGS, the amount and the quality of the DNA may, however, be variable, that may result in loss of sensitivity. (lines 221-222 and 245-250) Full names of variants detected are given in the legend to Figure 1.Reviewer 2 Report
For the study entitled “High genetic diversity and no evidence of clonal relation in synchronous thyroid carcinomas associated with Hashimoto’s thyroiditis – a next generation sequencing analysis”, authors collected 47 thyroid carcinoma and Hashimoto’s thyroiditis, after sequencing and statistical test, they detected significant differences with a wide spectrum of pathogenic gene variants. They also claimed that different BRAF status in coincident thyroid carcinoma foci within the same organ outlined a special challenge for molecular follow up and therapeutic decision. This study collected valuable samples. While, the results is too descriptive to emphasize any important information. In addition, several critical information did not present in manuscript. For example, “out of the 262 total TC cases evaluated 47 showed histological signs of HT (line 115)”, is any figure to show “histological signs” in manuscript? They collected 14 sample with both thyroid carcinoma and Hashimoto’s thyroiditis, which is an important recourse, while why they only sequencing 5 of them? Is there any important reason to do so? The sequencing data should be disposed on public data portal, e.g., GEO.
Minor comments:
What’s the x axis for Figure 2.
What the NGS sequencing for? Whole genome sequencing? Whole exome sequencing or others?
Please cite study for data analysis tools (line 105).
Author Response
Thank you for the supporting opinion and for the valuable remarks. Below please find our answers to the open issues and the modifications done:
Histological signs of HT: HT was histologically stated by the massive accumulation of lymphocytic infiltrate in the non-neoplastic thyroid parenchyma. The amount of the infiltrate is given Table 1. Clinical findings supportive HT were also provided for all 47 patients. Altogether 14 patients presented with at least two cancerous foci within the same surgical sample. In this first attempt 6 cases (6x2 tumor foci) could be selected where both foci were sufficient for NGS (quality and amount of FFPE sample or isolated DNA, measure of coverage). The upload of the data is planned in the near future in diverse open databases as usual NGS was done using a 67 gene panel covering the most relevant solid cancer related oncogenes and suppressor genes. Oncogenes included are listed in the Methods section (line 102-107). Accoring to our current experience WES would not allow a first glance conclusion at a much higher level of analytical complexity. A reference to the Archer sequencing tools was added in the Methods section (Ref. no. 17).Round 2
Reviewer 2 Report
The manuscript is improved. I have no further comment on the revised version.